# Thermal Energy Harvesting from Slow Variations in Environmental Temperatures

**DOI:** 10.3390/mi14061202

**Published:** 2023-06-06

**Authors:** Joshua Curry, Nick Harris, Neil White

**Affiliations:** School of Electronics and Computer Science, University of Southampton, Southampton SO17 1BJ, UK; nmw@ecs.soton.ac.uk

**Keywords:** energy harvesting, phase change materials, environmental sensing

## Abstract

With the Internet of Things expanding to more locations across our planet, power becomes the main factor affecting device longevity. There is a need for more novel energy harvesting systems that are able to power remote devices for sustained periods. This publication presents one such device. Based on a novel actuator that utilises off-the-shelf gas mixtures to generate a variable force from temperature change, this publication presents a device capable of generating up to 150mJ per diurnal temperature cycle; this is enough electrical energy to send up to three LoRaWAN transmissions per day using slowly changing environmental temperatures.

## 1. Introduction

Powering the Internet of Things is becoming an area of research with a greatly increasing need for practical solutions. As the demand for data increases in the modern age, solutions are required to power sensing devices in a variety of environments. Controlled environments such as inside buildings and residences can provide predictable energy sources such as temperature [1], airflow [2], light [3], and RF energy [4], which can be harvested and used to power sensing devices. Uncontrolled, outdoor environments, however, lack these predictable sources of energy, with constant changes in conditions and parameters resulting in a great difficulty in powering remote sensors for long periods. Energy harvesting systems which utilise photovoltaics hold the largest market share to this day for use in remote environments, but they are not without their drawbacks. These drawbacks include the requirement of regular cleaning to maintain efficiency and a potential to be limited in operation by the onset of vegetation [5]. There is a need for an environment-agnostic energy harvester that can make use of the fundamental variable factors present in most uncontrolled environments to support much longer-term deployments of sensing devices without the drawbacks of existing solutions.

This publication presents an energy harvesting system that utilises the phase change of gases to provide a variable force from which an electrical energy output can be extracted. Previous publications in this field have explored the performance of the actuator [6] and a proposed energy conversion system [7]. In this publication, the energy output of this energy harvesting system was optimised and its performance was verified in both controlled tests in an environmental chamber and uncontrolled tests in a real environment.

The findings of this publication show that this type of energy harvesting system can be used in a real-world environment to provide up to 150 mJ of energy per day, which provides enough energy for up to three transmissions of sensor data via LoRaWAN from a test device.

## 2. Background

With new connectivity solutions such as LTE-M and NB-IoT gradually making their way into the IoT market, the prospect of gathering data from remote locations for study and real-time analysis is gradually becoming more of a reality. However, there is a need to power these kinds of remote sensing systems for extended periods. Battery technology supported by a suitable energy-harvesting system, and, in most commercial cases, a photovoltaic system is starting to become the de facto standard for remote IoT sensing devices. However, this photovoltaic technology is not without its drawbacks, as it requires regular cleaning and maintenance to maintain efficiency.

Uncontrolled environments in which remote sensors are likely to operate are notoriously difficult to harvest energy from due to their often widely fluctuating environmental conditions [5]. The one environmental parameter that can be assured to change on a diurnal basis is temperature. Different regions of the world experience different changes in diurnal temperatures that are dependent on their distance from the equator, as primarily these temperature changes are driven by the angle of incidence of the sun in combination with local weather conditions such as cloud cover, humidity, and precipitation [8].

Energy harvesting from temperature changes has seen a significant amount of research in recent years, with new thermoelectric and pyroelectric solutions offering new and increasingly efficient ways to harvest energy from temperature gradients [9]. However, in many outdoor environments, temperature gradients cannot be guaranteed to exist all the time, and, with the exception of areas in direct sunlight, gradients of enough magnitude to power IoT devices are hard to find.

Thermal energy harvesting using phase changes provides a possible route to harvest the slow diurnal temperature change of an environment and use it to power an electrical device, and, in recent years, there have been a number of publications in this area.

Chloroethane-filled bellows provide a possible solution to harvest energy from this slow change in temperature. Having first found use in the Atmos clock, invented by Cornelius Drebbel in the 17th century [10], chloroethane-filled bellows provide a force and displacement that are proportional to the temperature of the environment in which they are stored, which was used to power clocks and other mechanical devices. It has been shown that these bellows can be used in conjunction with a gearbox arrangement to provide a potential source of energy for IoT devices [11,12]. Atmos clock bellows have also been utilised with small sprung energy harvesting modules to power a real-world IoT device [13], where in they provided up to 21 mJ of energy per cycle within the range of 5 ${}^{\circ }C$ to 25 ${}^{\circ }C$.

Chloroethane is not the only gas that can be utilised for this purpose, and recent studies have shown that butane can provide a similar change in force and displacement across a useful temperature range when constrained by an appropriate constant-force spring. In one study [14], this concept was used to power a small, light vehicle that could slowly move across a flat surface when exposed to changing environmental temperatures.

Communications for IoT devices that operate in remote environments are integral to their operation, which are often the highest energy consumer in embedded systems. A variety of solutions exist in this space, from carrier-supported, subscription-based solutions, such as NB-IoT and LTE-M [15], to free solutions, such as LoRaWAN [16]. These low-power, wide-area networks (LPWAs) disrupt the status quo of IoT communication technologies by providing a low data rate, as well as transaction-based approaches to data transfer that are well-suited for energy-constrained environments [17]. In order to usefully power an IoT device in an uncontrolled remote environment and compete with the large amount of energy available from photovoltaic solutions, any energy harvesting solution designed for such environments should be able to provide enough energy to power at least one of these technologies.

## 3. System Design

This publication builds upon a novel energy harvesting actuator that makes use of a double-acting pneumatic cylinder to provide variable force and displacement from the pressure difference between a propane–butane mixture and compressed air. This variable force and displacement output can be harnessed through appropriate gear reduction to drive a sprung release mechanism, which, upon a certain input force threshold being reached, releases its mechanical energy into an electrical generator. Figure 1 shows a block diagram describing this overall energy harvesting system.

The core of this system is the actuator, which is a double-acting pneumatic cylinder (SMC CDG1BN20-100Z) that allows the temperature-dependent pressure of a propane–butane gas mixture to be compared with the pressure of a standard cylinder of compressed air. As the temperature increases, the pressure of the propane–butane gas mixture increases at a higher rate compared that of the compressed air, thereby allowing the net force output from the cylinder to be positive or negative depending on temperature.

In previous published work [6], it was shown that this type of actuator design showed great potential for energy extraction from this difference in the behaviour of gases compared with air when exposed to a temperature change. It also showed a high degree of tuneability, with numerous design factors that could be adjusted to change the behaviour and output of the actuator.

Primarily, the concentration and mixtures of gases could be adjusted to change the performance of the actuator at different temperatures. These mixtures could also have their performance “muted” by reducing gas concentrations in the driving side of the cylinder and adding air to increase the maximum temperature to which the system can be exposed to without over-pressuring. Physical factors such as the size and bore of the cylinder can also be varied to change the amount of force developed by the actuator and the distance across which the force is applied, which can be used to greatly influence the size and performance of the system.

A “resistance pressure” can be set using the pressure of the fixed volume of compressed air to which the gas pressure is compared, which allows the temperature range across which cylinder expansion occurs to be adjusted. This allows the cylinder to be precisely tuned to expand and contract at different temperatures, meaning that it can be harmonised with a wide variety of different environments.

In further previous work, this actuator was integrated with an energy conversion system [7] consisting of a torsion spring and a 3D-printed release mechanism that allows the actuator to be tuned to release energy in a fast motion at a certain temperature. This meant that a single energy harvesting event could occur at a predetermined point as the diurnal temperature rose, thus resulting in the efficient extraction of potential energy into a generator. A simulation was developed for the release temperature of the system, and this was compared with experimental results to show that the system could release consistently at the same temperature across multiple cycles.

After the completion of these two works, the converter was shown to output 10 mJ per day of energy per release, and this was collected directly from stepper motor windings using an electrolytic capacitor and diode. Whilst this proved its functional operation, such an energy output is an order of magnitude away from powering an IoT device with LPWA capability, and the results show a scope for optimisation.

Figure 2 shows a functional diagram of the energy harvesting system explored in previous works. The expanding butane–propane mixture causes force $F_{A}$ to be developed on the cylinder ram. This is opposed by $F_{B}$ provided by a fixed volume of compressed air in an external cylinder. As the temperature increases and $F_{A}>F_{B}$, the cylinder ram begins to move, which winds the torsion spring, which opposes with torque $\tau_{S}$. The outer part of the torsion spring is held in place by a sprung latch, which is held against the spring body by force $F_{L}$. As the shaft rotates, a release cam affixed to it eventually makes contact with the sprung latch, and $F_{L}$ is overcome, thereby releasing the outer part of the torsion spring, which transfers the stored torque $\tau_{S}$ through a set of spur gears into a stepper motor. This stepper motor converts the released torque into electrical energy.

The elegance of this system is that the release of the torsion spring is achieved by overcoming a set level of input torque, and, therefore, it occurs automatically at a set temperature. As a limitation of its current mechanical design, this energy harvester only harvests energy when the input temperature increases. This is due to the mechanical orientation of the sprung release catch and restrictions in the design of the torsion spring, which currently can only store energy when its input is rotated in a single direction. However, in the future, this design could have the potential to harvest energy across both an increasing and decreasing temperature gradient with design alterations.

In its current state, this energy harvesting solution presents a sizeable footprint of approximately 40 × 40 × 20 cm. This is due to being implemented using mainly commercial off-the-shelf (COTS) components, which allow for the easy interchangeability of parts during its further development. Despite currently being implemented in the macroscale, its reliance on the core physical principles of phase change materials and simple mechanical concepts gives it great potential to be miniaturised into a significantly smaller custom solution once its functionality has been demonstrated and its device performance has been characterised.

### 3.1. Optimisation of Electrical Energy Harvesting System

Due to its low cogging torque and small footprint, a small bipolar stepper motor with part number 11HS20-0674s was selected to use as the electrical energy converter for the harvester. When stepper motors are used as generators, they output an AC voltage. Dependent on if the motor is bipolar or unipolar, either two or four alternating signals are output. In order to store this energy and use it to power a DC device, the first step of any energy harvesting system must be conversion to DC. Figure 3 shows a prototype circuit for converting the energy output of a single phase of the 11HS20 bipolar stepper motor into DC for capacitive storage. Datasheet values for winding resistance and inductance are included to facilitate further modelling.

In order to optimise the power transfer from the motor into the electrical energy storage element, the impedance of any harvesting circuit should be tuned to approximately match that of the generator. In order to do this and maximise the power transfer into the energy storage element, there are two components that need to be carefully selected. The first of these, $C_{store}$, represents the size of the energy storage element. The energy input to the conversion system is in the form of a limited short burst of energy for a fixed time. This means that the optimal value of $C_{store}$ will need to provide a good impedance match for as much of the energy burst as possible.

Another parameter that requires careful selection and optimisation is $C_{series}$, which is placed in series with the stepper motor winding to cancel inductive losses. This component forms an RLC filter with the series resistance and series inductance of the winding, and, as such, it needs to be adequately specified as dependent on the frequency of the AC signal output by the stepper motor during a release of the conversion system.

To facilitate the optimisation of these values, the stepper motor was first configured with one phase connected through a simple rectifier to a large capacitor, as in Figure 3, whilst the other phase was connected to an oscilloscope. The AC output of the open phase of the stepper motor was captured and saved to a disk, which was then used as an input for SPICE simulation.

By utilising the measured open circuit voltage of one of the stepper motor windings, LTSpice [18] was configured to perform sweeps of both capacitor values to obtain an approximate match for their optimal values. Figure 4 shows the output of these simulations, showing an optimal power output of 107 mJ for a single phase, with $C_{store}$ equal to 2500 μF and $C_{series}$ equal to 3.5 μF. After the release, the maximum $C_{store}$ voltage was 9.23 V.

### 3.2. Integration with a Low-Power Sensing Device

In order to extract the maximum amount of energy efficiently from this system, two sets of rectifiers and storage capacitors can be placed in series. This brings with it advantages, such as the ability to utilise more of the energy stored before any regulation system reaches its minimum operating voltage. It also means that the impedance of the match of the harvester with each phase is not affected, as the phases are not connected in parallel. A parallel configuration could be created by utilising diodes to combine the two energy stores; however, this would result in an undesirable extra voltage drop across the combining diodes, which would affect the system efficiency.

Figure 5 shows the final optimised circuit design used for testing. The two phases of the stepper motor were placed in series, and the input was fed into an LTC3129-1 buck/boost converter [19], which was configured to output 3.3 V. Values close to those identified as optimal by the SPICE simulation above were used for $C_{store}$ and $C_{series}$, which were implemented as 2200 μF and 4.7 μF, respectively.

In order to facilitate the testing of this energy harvesting circuit under real-world conditions an example LoRaWAN device was used, comprising of a TPL5111 ultra-low-power system timer [20], Atmel ATSAMD21G18A, and a HopeRF RFM95W LoRa Radio.

Due to the relatively small $C_{store}$ capacitance, any powered system needs to be designed to have a minimal quiescent current when not in operation so that the voltage stored in the capacitor is only used when the device is performing its mission task. Due to the ultra-low (35 nA) quiescent current of the system timer used in this design, the capacitor voltage can be maintained for longer periods between transmissions. Alongside this, the use of an SPI-Driven LoRa radio gives finer control over power usage when compared with similar LoRa radio solutions that utilise AT commands over UART and contain their own microcontrollers.

This device was fabricated, and its current usage to send a single LoRa message was analysed using a Nordic semiconductor power profiler kit. Figure 6 shows the current consumption of a single 8-byte data transmission from a temperature/humidity sensor onboard the test device over LoRaWAN at a low radio transmission power. This trace indicates that each sensor reading and LoRaWAN transmission event at low-transmission power utilised an average current of 27.60 mA across a period of approximately 100 ms. This equates to 4.36 mC of charge usage from a capacitive storage element.

## 4. Methodology

In order to characterise the performance of this energy harvesting system, the performance of the energy harvester needed to be analysed under a variety of conditions. For controlled testing, the experimental system was placed in a Weiss WKL 100 environmental chamber and subjected to a fixed, repeatable temperature profile to provide a slow increase in spring tension until the release point. For uncontrolled testing, the device was placed in an outdoor location.

### 4.1. Performance of Energy Harvester under Controlled Conditions

Figure 7 shows the experimental configuration in the environmental chamber. In the centre of the left image, the double-acting pneumatic cylinder can be observed with a toothed gear rack attached to the cylinder ram, which exhibits movement across the temperature cycle. The right hand side of the cylinder was filled with 3 g of Rothenberger Super 100 propane/butane mixture, and the left hand side of the cylinder was connected to a 1500 cm${}^{3}$ compressed air tank.

The compressed air tank was prepressurised to 4.25 bar to provide a suitable ‘resistance pressure’ to cause the actuator to move and an energy harvesting event to occur at approximately 30 ${}^{\circ }C$. This pressure was selected by utilising the empirical results obtained as part of previous work [7].

At the top left of the image, the 3D-Printed converter assembly can be seen, consisting of a clock spring, release latch, and release cam which rotates about the shaft and releases the latch when the cylinder ram has moved by a set amount. In the top right of the image, the circuit shown in Figure 5 can be seen, consisting of 1N5817 Schottky rectifier diodes with a maximum forward voltage of 450 mV.

In order to study the performance and reliability of the short energy harvesting event intrinsic to this harvester, a temperature profile was designed to allow for a reset of the release latch by cooling the device to a low temperature and then restoring it to a temperature just below the release point. A slow temperature rise was then implemented until the device released and the consistency of its operation could be studied.

In order to achieve this, the device was subjected to a slow temperature profile varying between 0 ${}^{\circ }C$ and 36 ${}^{\circ }C$, and its operation was studied. For the first part of the experiment, the temperature was lowered to 0 ${}^{\circ }C$ for 20 min to allow the device to cool and to ensure a reset of the converter latch. The temperature was then raised at a rate of 1 ${}^{\circ }C$/min^−1^ to 30 ${}^{\circ }C$, where it was held for another 20 min. When the device was equalised at this temperature, the temperature was then raised again at a slower rate of 0.1 ${}^{\circ }C$/min^−1^ to 36 ${}^{\circ }C$, where, at that point, the experiment ended. After a short 20 min delay at this highest temperature, the experiment was then cooled to 0 ${}^{\circ }C$ at a rate of 0.5 ${}^{\circ }C$/min^−1^ to allow it to reset once more. The experiment was then looped to allow reliability to be explored.

A number of metrics were collected throughout the experiment. The temperature of the double-acting pneumatic cylinder was measured by a Honeywell HIH6120 temperature sensor read by an Arduino MKRWAN1300. The gas pressures in both chambers within the cylinder were monitored using Honeywell PX3AN2BS250PAAAX absolute gas pressure sensors. All sensing devices used for this experiment were sourced from RS Components UK.

In the rightmost image of Figure 7, the experiment can also be seen outfitted with AprilTag fiducial markers [21]. These markers allow for very precise positioning of objects within an image, and, in this case, facilitate monitoring of the displacement of the cylinder throughout the experiment duration.

### 4.2. Performance of Energy Harvester under Uncontrolled Conditions

In Figure 8, the same generator was placed in an outdoor location. A number of days in late March in a location in the southern United Kingdom were selected for the outdoor experiment due to the relatively low night temperatures and moderate daytime temperatures with some incident sunlight.

To keep wet weather conditions from affecting system monitoring, a simple plastic container was used to cover the experiment. This container allowed for the protection of system components but also acted as a tuning factor, thus insulating the device from the surrounding environment. This allowed the device to equalise with the environmental temperature at night when it was cooled for a significant period and for the temperature of the device to be much higher during the day than the air temperature due to the effect of incident sunlight on the insulated air volume.

The experiment was configured to record the same metrics as the previous experiment above, with the exception of cylinder displacement, due to its relative complexity to configure and ability to be inferred from other data.

## 5. Results and Discussion

### 5.1. Performance of Energy Harvester under Controlled Conditions

Figure 9 and Figure 10 show the results of the experiment under repeatable controlled conditions. In Figure 9, the system displacement was analysed by plotting the position of the AprilTag fiducial markers over time. System temperature was provided as a comparison metric. In Figure 10, the absolute pressures of both sides of the double-acting pneumatic cylinder can be observed alongside the generated voltage.

The data in these figures provide insight into the operational characteristics of the energy harvester. Primarily, displacement can be seen to be out of phase with temperature due to the time taken for thermal energy from the chamber air to transfer to the apparatus. In the figure, the displacement of the cylinder can be seen to have risen as soon as $P_{gas}$ exceeded $P_{res}$, as expected. This start of movement also caused a change in the pressure of the cylinder at t=2 h, which was likely due to the increase in volume causing a momentary drop in pressure. The pressure and displacement continued to increase at a slower rate due to the decrease in the rate of the input temperature profile described above. Eventually, at T= 33 ${}^{\circ }C$, the displacement changed by a large amount as the release cam overcame the sprung latch, the torsion spring was released, and electrical energy was generated.

At this release point, a voltage of 16.55 V was generated in the storage capacitor, with an overall capacitance of 1100 μF. By utilising the simple formula E=0.5 CV^2^, an energy value of 150 mJ is obtained as the amount of energy harvested into $C_{store}$.

In real-world operation with the example LoRaWAN device described above, not all of this energy would be able to be used by the system due to the limitations of voltage regulation. The example system described above utilised 4.36 mC of charge per transmission and had a very low leakage current, which could be considered negligible compared to the intrinsic leakage of the capacitor. With a maximum voltage of 16.55 V generated by a release, utilising Q=CV, the charge stored by the capacitor was equal to 18.20 mC. The charge stored in the capacitor at the minimum startup voltage of the boost converter of 1.92 V is equal to 2.11 mC. This gives a usable charge of 16.09 mC, which means that a transmission consuming 4.36 mC can theoretically occur a maximum of three times.

When tested with the example LoRaWAN device above, this number of transmissions was successfully achieved with a delay of 1 min between transmissions. Longer periods resulted in less successful transmissions due to the intrinsic leakage of the storage capacitors and the quiescent current of the boost converter.

### 5.2. Performance of Energy Harvester under Uncontrolled Conditions

Figure 11 and Figure 12 show the performance of the energy harvester during outdoor testing. A single day of data in the period of 8 April 2023 to 9 April 2023 is displayed. Figure 11 shows the temperature of the device during the outdoor test, and Figure 12 shows the pressures of both sides of the double-acting pneumatic cylinder alongside generated voltage.

This real-world test of the energy harvesting system shows that it could operate in uncontrolled conditions in a similar fashion to its operation under controlled conditions. As shown in the figure, the energy harvesting system released at a temperature of 32 ${}^{\circ }C$, generating 15.6 V in the storage capacitor, approximately matched the previous experiment within a margin of error.

Reasons for the variation in release temperature and pressure are likely related to the fact that the resistance pressure vessel was subjected to fixed temperatures in one test and variable temperatures in another, which caused the resistance pressure to deviate from its fill pressure. In controlled testing, it was kept at a constant 25 ${}^{\circ }C$, whereas, during uncontrolled testing, its temperature varied with the temperature of the environment. This caused the pressure to drop from the fill pressure of 4.25 bar as the environmental temperature decreased after filling.

## 6. Conclusions

Both of the experiments discussed by this publication have shown that a novel energy harvesting technology, based on differences in pressure between gas mixtures and compressed air, shows great potential for energy harvesting for IoT systems in real-world environments.

In both controlled and uncontrolled testing, this energy harvester generated 150 mJ of energy per diurnal temperature cycle, which is enough to transmit three LoRaWAN messages containing sensor data. This brings this technology closer to being able to compete with existing energy harvesters for remote sensing solutions by supplying a useful alternative energy source, which can be used for low-duty-cycle data transmissions.

In the future, further testing will need to be performed to verify the operation of this type of device across longer periods of time and during seasonal changes, which affect the thermal gradient across which the system operates. Further integration of the system will be conducted, thereby harmonising the system with an appropriate long-term energy storage element to allow it to operate an IoT device for longer periods between energy harvesting events.

In its current implementation, the energy harvester operates over a wide thermal gradient and only harvests energy on the outward stroke of the actuator. Mechanical optimisation will be investigated to trigger the harvesting of energy on both heating and cooling cycles to further increase the system energy output. In addition to this, more research will be conducted into exploiting the thermal properties of the gas mixture to provide larger, more favourable responses across smaller thermal gradients.

## Figures and Tables

**Figure 1 micromachines-14-01202-f001:**
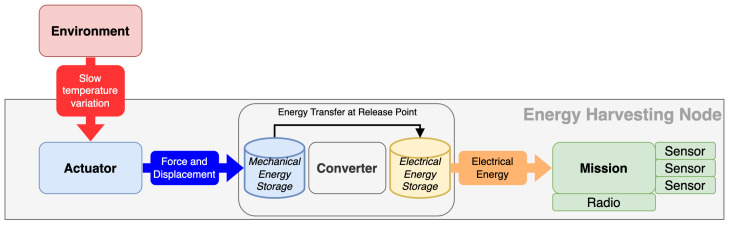
Block diagram of proposed energy harvesting system.

**Figure 2 micromachines-14-01202-f002:**
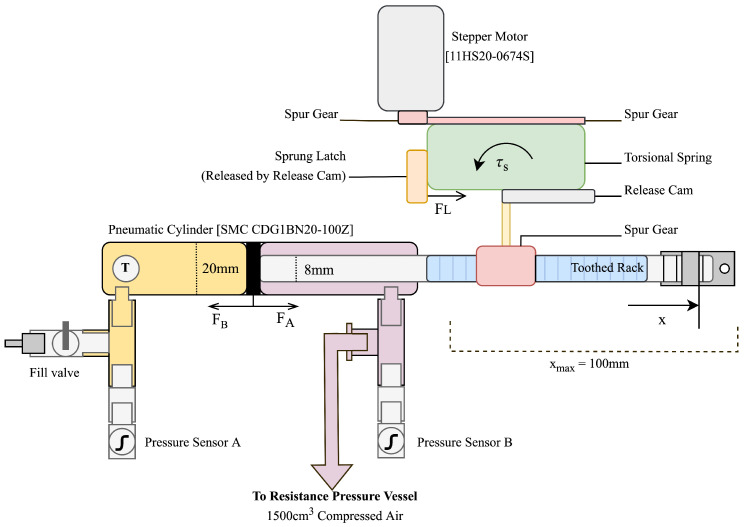
Functional diagram of energy harvesting system.

**Figure 3 micromachines-14-01202-f003:**
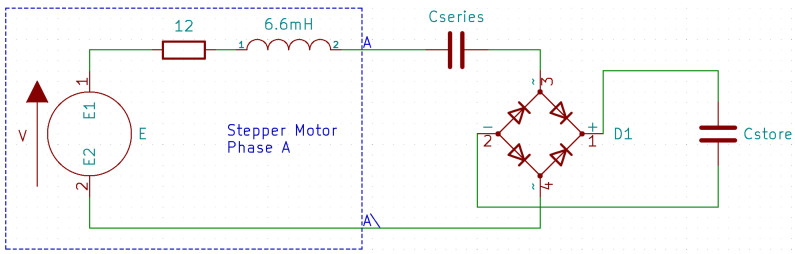
Design of rectifier stage for single phase.

**Figure 4 micromachines-14-01202-f004:**
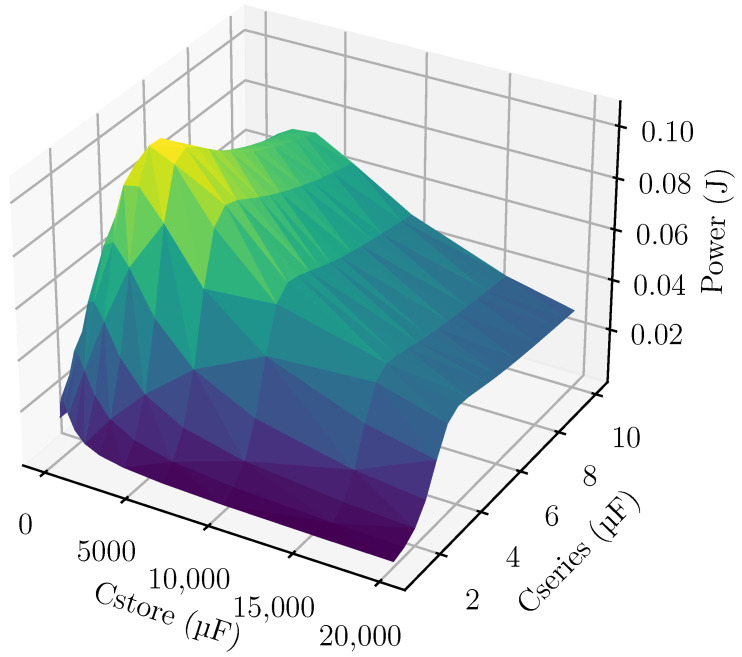
A plot showing the optimal values for $C_{store}$ and $C_{series}$ from SPICE simulation.

**Figure 5 micromachines-14-01202-f005:**
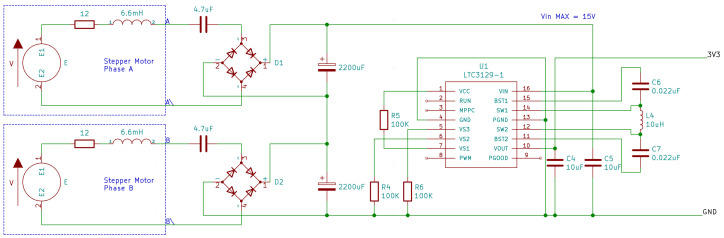
Integrated design of energy harvester circuit with regulator.

**Figure 6 micromachines-14-01202-f006:**
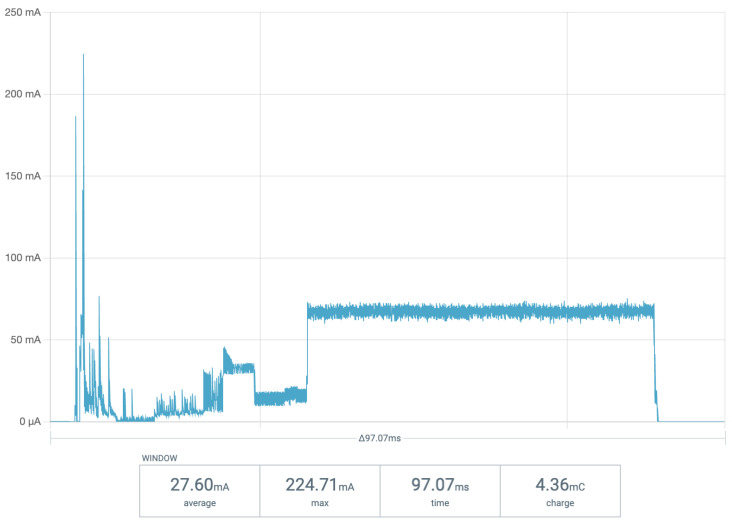
Current consumption of test device during transmission event.

**Figure 7 micromachines-14-01202-f007:**
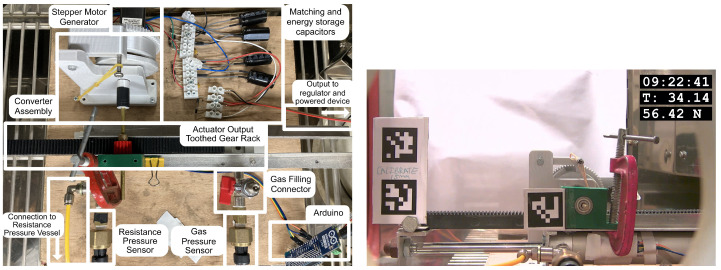
Experiment configuration in environmental chamber (**left**) and displacement measurement (**right**).

**Figure 8 micromachines-14-01202-f008:**
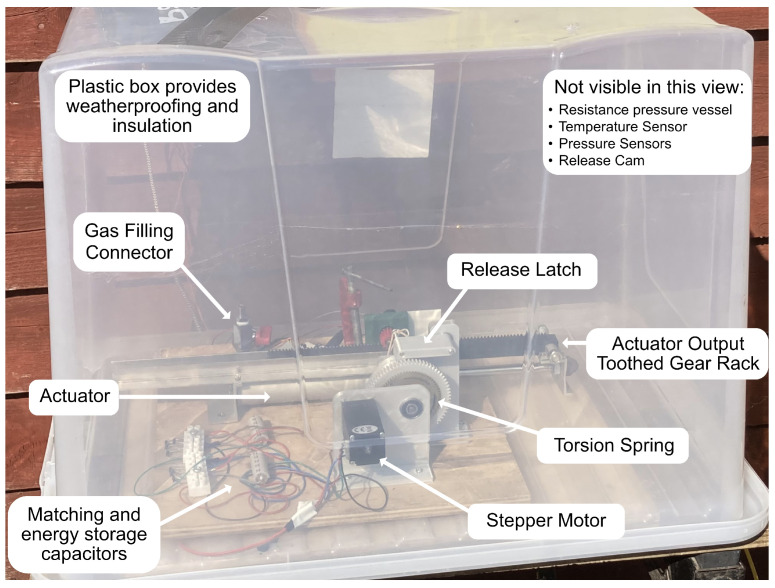
Configuration of outdoor experiment.

**Figure 9 micromachines-14-01202-f009:**
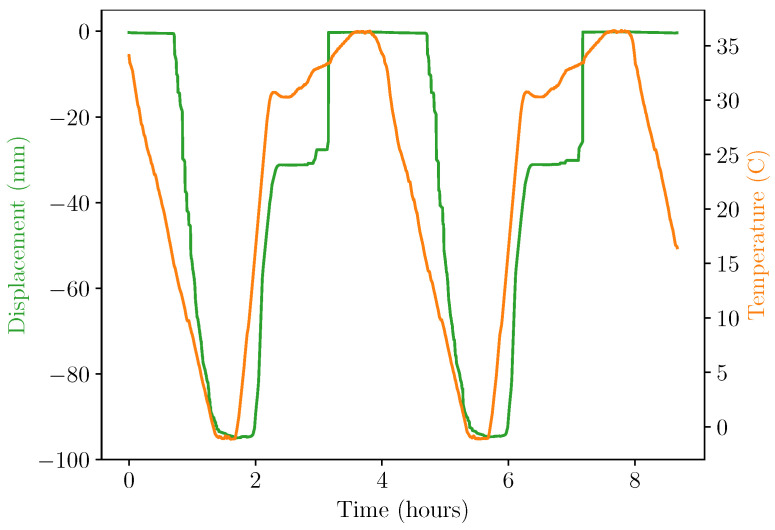
Experiment temperature and displacement during environmental chamber test.

**Figure 10 micromachines-14-01202-f010:**
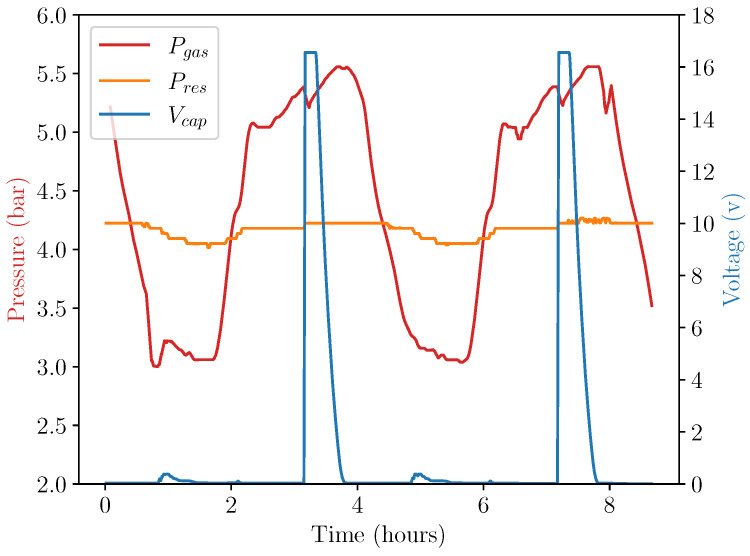
Device performance during environmental chamber test.

**Figure 11 micromachines-14-01202-f011:**
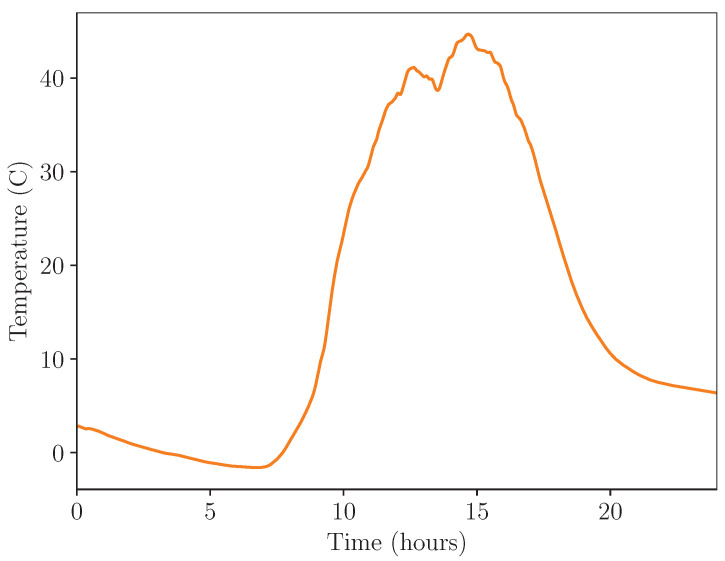
Experiment temperature during outdoor test.

**Figure 12 micromachines-14-01202-f012:**
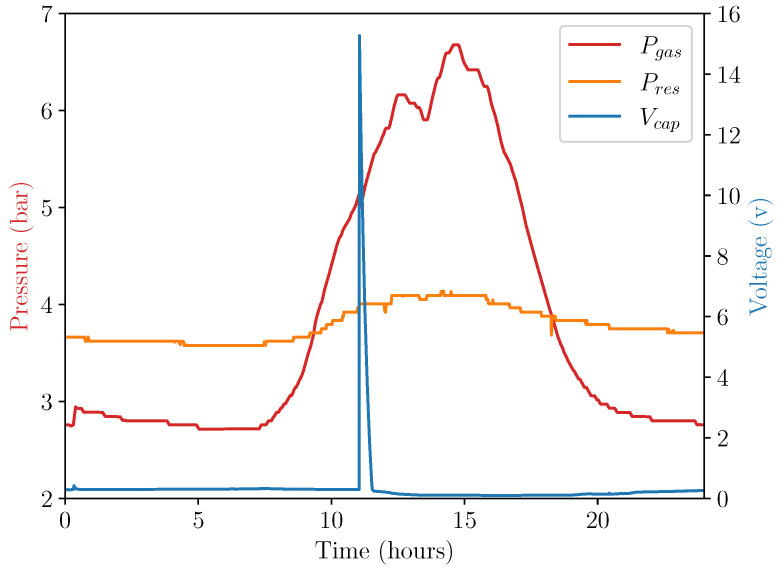
Device performance during outdoor test.

## Data Availability

The data presented in this study are available on request from the corresponding author.

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
