# Peer review of "Thermal Energy Harvesting from Slow Variations in Environmental Temperatures"

_micromachines, 2023, doi:10.3390/mi14061202_

Round 1

Reviewer 1 Report

1.     General remark

First of all, I would like to congratulate the Authors for conducting this research. The submitted manuscript describes interesting energy harvesting technique that is capable to generate electricity from natural air diurnal temperature fluctuations. Such an energy generator is of very high importance and value regarding huge need to replace conventional supply using batteries or wires. Temperature fluctuations are very advantageous source of energy owing to their omnipresence. Thus such energy harvesting technique can be used almost everywhere which is an asset modern electronics strongly desires. Research presented in the submitted manuscript makes a step towards achieving energy-autonomous electronic devices and thus has strategic importance for the scientific community.

Authors are proposing an extension of C. Drebbel’s idea allowing to produce electricity from natural air temperature fluctuations. In the submitted material the Authors are presenting full chain of device development: (i) idea; (ii) modelling & simulations aiming at performance boost; (iii) practical realization and (iv) characterization. The device was experimentally characterized in laboratory and external conditions (beginning of April 2023). The reported energy per day is 150mJ which is sufficient to send up to 3 messages via LoRaWAN. The experiment confirms the feasibility, reported energy is high enough to totally supply numerous electronic devices.

In light of the aforementioned arguments, I recommend to the Editor accepting this manuscript for the publication in MDPI Micromachines under major revision. I have few questions and suggestions to the Authors that are listed in the following.

2.     Suggestions, remarks and questions

a.       Question #1 – Is the energy released only once per day? It is possible to release the energy more than once per day if the temperature conditions are favourable?

b.      Fig. 7 (left) – I suggest placing total width and length dimensions for the system.

c.       Question #2 – the characterized harvesting device depicted in Fig.7 (left) intuitively seems quite big and cumbersome. I would like to know if there is any possibility to realize such harvester in miniaturized form. Size is very important for the modern electronics. The demonstrator described in the submitted manuscript is much bigger than the battery. Could Authors describe the miniaturization possibilities in order to improve the attractiveness of this device for the industry.

d.      Fig. 4 or Lines 169-171– Please double check the Fig.4 or text in lines 169-171. To my eyes there is a mismatch between the figure and text;

e.       Fig. 4 – Z-axis label is missing please add

f.       Lines 61-64 – The Authors wrote: “First finding use in the Atmos clock, invented by Cornelius Drebbel in the 18th century [10], […]” please double check the first realization of C. Drebbel. C. Drebbel died in 1633, so making the firs realization in 18th century is doubtful. Please double check this information.

g.       Question #3 – Could You please explain how the latch reset is realized? This requires human intervention?

h.      Line 237 – typeset error there is “ Figure 7 7”;

i.        Figure 9 – in the Fig. 9right for time t1h there is small and short increase of Vcap this is repeated for t5h and probably would also repeat for t9h (but it is out of experimental scope). Could You please explain why such Vcap increase when Pgas and temperature are decreasing? Moreover, such Vcap increase is observed only for the experiments in the climate chamber but are not visible in the outdoor experiment – why? The temperature difference in the outdoor experiment was higher than in the climate chamber.

j.        Figure 9 – I suggest replacing side-by-side configuration. Placing one figure above the other will be much easier for the Reader. It will be easier to link displacement/temperature with pressures/voltage for a given time;

k.      Figure 10 - I suggest replacing side-by-side configuration. Placing one figure above the other will be much easier for the Reader. It will be easier to link displacement/temperature with pressures/voltage for a given time;

l.        Figure 10 – the diurnal temperature difference is DT40-45°C, which intuitively for me seems a lot for south UK at the beginning of April? Could You please comment and justify the temperature measurement;

Author Response

Dear Reviewer,

Many thanks for taking the time to review our manuscript, "Thermal energy harvesting from slow variations in environmental temperatures". We have taken your useful comments onboard and made a number of changes to the manuscript. Please find a list of these changes below.

- More detail has been added about how the current iteration releases energy once per day, but has the potential for mechanical modification so that energy can also be extracted when the system cools.

- A section has been added on the miniaturisation potential for the device demonstrated in this publication. The current iteration is very large compared to other battery-based solutions for powering modern devices, but definitely has the potential to be miniaturised.

- Figure 4 was adjusted for clarity

- An oversight here placed C.Drebbel in the 18th instead of 17th century, the text has now been adjusted to correct this.

- The latch reset occurs automatically using a spring, and some clarification has been added to the text to indicate how this occurs.

- In description Figure 9, an explanation has been proposed for the small increase in voltage during the cooling cycle.

- Line 237 typesetting error fixed. 

- Figure 9/10 moved from side-by-side configuration to individual images.

- Clarification of the observed dT in the outdoor test has been added. The larger diurnal gradient was due to the addition of the plastic box over the experiment and incident sunlight heating the air inside. 

Reviewer 2 Report

The paper is incremental research from references 6 and 7, which the authors correctly stated in the Introduction. The scientific contribution is small, but there is an engineering one. Hands-on demonstration of the complete system in different environments is beneficial. Some corrections are required before publication:

Figure 3 is redundant because it is repeated in Fig. 5. Additionally, it is desirable to check the connection of the rectifier bridge.

A label is missing on the z-axis in Fig. 4.

In the rightmost image of Figure 7 7 ... - the number 7 appears twice.

Utilizing the simple formula C = 0.5CV2 ... - the formula should be corrected.

Author Response

Dear Reviewer,

Many thanks for taking the time to review our manuscript, "Thermal energy harvesting from slow variations in environmental temperatures". We have taken your useful comments onboard and made a number of changes to the manuscript. Please find a list of these changes below.

Figures 3, 4 and 5 have been re-rendered to reflect suggested changes and fix issues found by reviewers.

The reference link was fixed in "In the rightmost image of Figure 7". 

The formula "E = 0.5CV2" was corrected.

Reviewer 3 Report

Review on: Thermal energy harvesting from slow variations in environmental temperatures (micromachines - 2380687)

The manuscript presents a thermal energy harvesting system with focus on the optimization of the energy extraction to maximize energy output. The device is verified in both a controlled test environment and outside in a real environment.

Here is a list of points for improvement of the paper:

In order to maximize the energy output of the harvesting system several system components require optimization. In this paper, the focus is on the circuitry. However, the reader may wonder about other components such as the generator and the spring mechanism. Could the authors provide some information about the choice of these other components? Are they designed / dimensioned together with the pneumatic system components? A simple hypothesis may be, that the energy output is increased much further just by the use of a different generator.

Figure 2: please review the direction of FA and FB. According to the text, the butan-propane mixture is in the left chamber of the pneumatic cylinder. When expanding, the piston is supposed to move towards the toothed rack (x-direction)?

At present state, energy can be extracted at the “rising edge” when the temperature is increased from 0°C to 36°C. Would it be possible to exploit also the “falling edge”, i.e. to extract energy when the temperature is falling and the piston moving in opposite direction?

The pre-pressure of the compressed air tank is set to 4.25 bar. Is there a particular reason for this value? Is it to adjust the release temperature?

Figure 9: please improve it for better readability without color (e.g. arrows to indicate which curve is displacement or temperature)

Figure 9: at about 2.5 hours, the piston stops to move despite Pgas > Pres. What is the reason for this? Is the opposing force by the torsional spring and FB equal to FA?

Conclusion section: what kind of applications are possible with the energy harvesting system presented? How does the system compare to other solutions, e.g. photovoltaics (number of components, size, power / energy output)? What are the advantages of the presented system over other solutions? Some comments with respect to scaling of the system would also be helpful.

Author Response

Dear Reviewer,

Many thanks for taking the time to review our manuscript, "Thermal energy harvesting from slow variations in environmental temperatures". We have taken your useful comments onboard and made a number of changes to the manuscript. Please find a list of these changes below.

More clarification has been provided about the choice of other mechanical system components as requested.

Figure 2 was adjusted so that forces F_A and F_B aligned with their description in the text.

The current iteration of this system design only harvests energy on the "rising edge" of environmental temperature. It could be adapted to also harvest energy on the "falling edge" and information clarifying this has been added to the future work section.

Reasoning added for pressure of resistance vessel being set to 4.25bar. This pressure adjusts the temperature of release of the system (i.e. the temperature at which the force of expansion overcomes the sprung latch).

At t=2.5hrs into Figure 9, the cylinder does indeed stop moving due to an equilibrium between the force of (tau_S and F_B) and F_A. Clarification has been added to the text to identify this phenomena and suggest reasoning for it.

Extra information added to future work section clarifying prospects of miniaturisation and presenting a short comparision of the energy harvester in its current state and other existing energy harvesting solutions.

Reviewer 4 Report

This paper describes a design of energy harvester based on differences in pressure between gas mixtures and compressed air from compressed air, and its prototype showing its capability to generate energy to power LoRaWAN transmsisions in both controlled and uncontrolled temeprature changes. This is an interesting concept that could be used in certain applications. The design, experimental details and results are described well. However, the paper can be improved in these areas:

1. In Background, it would be useful to describe the potential applications of this design of energy harvester, e.g., in what possible scenarios could this design to be used, where other energy harvesting sources are not available while there is a temperature change.

2. What is the originality in the mechanical design of this energy harvester? Figure 2 show the system design but it is stated that is from previous work.

3. Line 223 to 231, it would be useful to draw a temperature profile to describe these changes. Also an explanation is needed on how this profile is designed, also in what case scenarios would these changes possibly happen.

3. In what sort of temperature change this sytem is functional, i.e., dose it operate in a smaller temperature change than those in the controlled and uncontrolled temperature changes in these experiments.

4. Line 253 to 257, would be useful to have a figure comparing the temperature change to the pressure change and voltage ouput etc., like in the Figure 10.

5. Figure 279, the system only release energy in the end of the temperature cycle and can potentilly power 3 transmissions. However, what happen if the 3 transmissions need to be distributed during the day? Its function is limited if it can only power 3 transmissions within a short period of time. Instead, an energy storage should be considered so the transmissions can have intervals in between and can be powered by the energy harvesting system. 

Author Response

Dear Reviewer,

Many thanks for taking the time to review our manuscript, "Thermal energy harvesting from slow variations in environmental temperatures". We have taken your useful comments onboard and made a number of changes to the manuscript. Please find a list of these changes below.

Wording clarified in background section detailing suggested environments where this type of energy harvester could be useful.

The mechanical design of this energy harvester (actuator and converter/release latch assembly) is from previous work, however this publication covers the extension of this work and optimisation of the electrical part of the energy harvesting system. 

Clarification added to the design of the temperature profile to which the experiment was exposed. 

Energy storage and distribution of transmissions over a longer time period added to future work section.

Round 2

Reviewer 1 Report

I recommend publication of this article in the Micromachines. All my remarks have been considered and questions answered.

ATTENTION: There is a problem with the references in the manuscript. All in-text citations are "[?]" with no consecutive number labels. Moreover, there is no list of references at the end of the manuscript. I suppose that it is a small problem with the references plug-in/software and will be rapidly solved by the Authors.